# Unveiling Genomic Islands Hosting Antibiotic Resistance Genes and Virulence Genes in Foodborne Multidrug-Resistant Patho-Genic *Proteus vulgaris*

**DOI:** 10.3390/biology14070858

**Published:** 2025-07-15

**Authors:** Hongyang Zhang, Tao Wu, Haihua Ruan

**Affiliations:** Tianjin Key Laboratory of Food Biotechnology, School of Biotechnology and Food Science, Tianjin University of Commerce, Tianjin 300134, China; zhyang@tjcu.edu.cn (H.Z.); wutao@tjcu.edu.cn (T.W.)

**Keywords:** *Proteus vulgaris*, genomic island, horizontal gene transfer, antibiotic resistance genes, virulence genes

## Abstract

*Proteus vulgaris* is a foodborne pathogen commonly found in seafood, particularly shrimp, that can cause serious infections in humans. Of particular concern are multidrug-resistant strains like P3M, which was isolated from farmed shrimp and poses significant treatment challenges. Our study examined the genetic mechanisms behind this bacterium’s antimicrobial resistance and disease-causing potential. Through genomic analysis, we identified specialized DNA segments called genomic islands that harbor both antibiotic resistance genes and virulence factors. These mobile genetic elements can transfer between bacteria, rapidly spreading dangerous traits. Additionally, we detected clusters of resistance genes associated with highly mobile DNA sequences that may accelerate their dissemination. Understanding these genetic transmission pathways is essential for developing effective strategies to control infections and safeguard food supplies. Our findings emphasize the urgent need for improved antibiotic stewardship in aquaculture systems to curb the spread of resistant bacteria. This work offers important guidance for researchers and public health officials striving to enhance food safety and protect community health.

## 1. Introduction

Foodborne pathogens, which include notable representatives such as *Salmonella*, *Escherichia coli*, *Proteus* spp., *Staphylococcus aureus* [1,2,3,4], are ubiquitous in human food systems and pose significant public health risks through their transmission via contaminated food, causing illnesses ranging from diarrhea to severe food poisoning [5]. Recent advances in molecular biology have led to the development of innovative detection techniques and analytical tools, which have substantially improved our ability to track transmission pathways, identify infection sources, and elucidate pathogenic mechanisms. These technological developments have consequently enhanced the diagnostic accuracy, detection sensitivity, and overall understanding of foodborne diseases [6,7,8].

The escalating use of antibiotics has driven a concerning rise in antimicrobial resistance (AMR) among foodborne pathogens [9,10]. This resistance mechanism functions as a biological defense system, effectively neutralizing antibiotic activity and promoting pathogen survival and proliferation [11]. Notably, the convergence of virulence factors and AMR mechanisms equips pathogens with enhanced capacity to overcome host defenses and complicates food safety interventions, thereby posing unprecedented challenges to public health regulation [12]. Consequently, comprehensive investigations into the molecular basis of pathogenicity and resistance are critical for developing effective food safety protocols, optimizing antibiotic stewardship, and designing targeted control measures.

The genomic repertoire of foodborne pathogens contains antibiotic resistance genes (ARGs) and virulence determinants, which collectively drive their pathogenic potential and drug resistance [13,14]. Horizontal gene transfer (HGT) serves as the primary vehicle for disseminating these traits, enabling rapid genetic adaptation through the acquisition of exogenous DNA [15]. Genomic islands (GIs), which are mobile DNA segments frequently harboring ARGs and virulence genes, have been extensively documented as major HGT substrates [16,17]. These modular genetic elements can integrate into microbial chromosomes and facilitate interspecies gene exchange, thereby expanding functional diversity and evolutionary potential [18,19,20]. Complementing GIs, insertion sequences (ISs) constitute another class of mobile genetic elements (MGEs) that orchestrate DNA rearrangements and accelerate trait propagation [21,22]. Collectively, these HGT mechanisms underpin bacterial evolution, including the emergence of enhanced virulence and multidrug resistance profiles.

*Proteus vulgaris* is a ubiquitous environmental bacterium that readily disseminates through food chains, with increasing prevalence as an emerging multidrug-resistant pathogen in aquaculture systems [23,24,25]. Clinically, this organism is a well-documented etiological agent of gastroenteritis, urinary tract infections, and other opportunistic diseases, establishing its role as a priority foodborne pathogen [26,27]. Notably, *Proteus vulgaris* exhibits concerning resistance phenotypes, including multidrug resistance (MDR) and intrinsic resistance to polymyxins (last-resort antibiotics), which severely limits therapeutic options for associated infections [28,29]. This resistance profile is especially alarming in aquaculture settings, where frequent gene exchange between environmental and clinical strains may accelerate the spread of resistance determinants [30,31,32,33].

While GIs have been well characterized as key vectors for AMR and virulence gene dissemination in other Enterobacterales like *E. coli* and *Salmonella* [17,34,35,36], their specific roles in mediating ARGs and virulence genes transfer in *Proteus vulgaris* remain unexplored, particularly within foodborne transmission chains. This study aims to characterize the GI repertoire of the multidrug-resistant *Proteus vulgaris* strain P3M from aquaculture sources, with a particular focus on assessing evolutionary conservation of resistance/virulence-associated GIs across 13 sequenced *Proteus vulgaris* strains through comparative genomics. Our focus on P3M’s GI architecture is motivated by its clinical resistance profile (resistant to seven antibiotic classes) [28], and the absence of prior systematic GI analyses in foodborne *Proteus vulgaris*, addressing critical knowledge gaps in understanding how this pathogen acquires and disseminates resistance and virulence traits under the One Health framework.

## 2. Materials and Methods

### 2.1. Proteus Vulgaris Genomes and Data Acquisition

The complete genome sequence of *Proteus vulgaris* strain P3M (accession: CP060211), along with 12 additional sequenced *Proteus vulgaris* genomes were retrieved from NCBI Nucleotide database (https://www.ncbi.nlm.nih.gov/nuccore, accessed on 11 April 2025) in GenBank (GBK) format. The analyzed strains included the following: *Proteus vulgaris* strain CCU063 (CP032663), FADDRGOS_366 (CP150645), FADDR-GOS_566 (CP033736), FADDRGOS_1507 (CP083628), HH17 (CP054157), LC-693 (CP063314), PvSC3 (CP034668), TAF3 (CP126335), USDA-ARS-USMARC-49741 (CP104121), ZN3 (CP047344), Ld01 (CP090064), and 2023JQ-00005 (CP137920). All genomes represented complete, annotated assemblies, with sequencing platforms, library protocols, and assembly methods detailed in their original NCBI submissions (accessible via the provided accession numbers). Detailed information of all these 13 *Proteus vulgaris* strains is listed in Appendix A.

### 2.2. Prediction and Comparison of Genomic Islands (GIs)

The online tool IslandViewer 4 (v4.0.1) was employed to predict the potential genomic islands (https://www.pathogenomics.sfu.ca/islandviewer/browse/, accessed on 15 April 2025) [37]. IslandViewer 4 integrates three distinct algorithms (SIGI-HMM, IslandPick, and IslandPath-DIMOB) to compensate for limitations of individual methods, achieving >90% consensus accuracy for GI boundaries. Among these, SIGI-HMM analyzes codon usage bias to identify potential genomic islands, while Island-Path-DIMOB assesses dinucleotide deviation for the same purpose. The IslandPick method employs genome-wide sequencing to detect large DNA fragments that exist independently within a genome but are absent in related genomes. Additionally, a comparative analysis of genome islands was conducted using the IslandCompare online tool (v1.1.0) [38] (https://islandcompare.ca/analysis?id=ff959500-555c-11ef-b9c1-059f1c7f7fe4, accessed on 15 April 2025). IslandCompare was employed because it uniquely visualizes GI conservation patterns across all 13 sequenced *Proteus vulgaris* strains, enabling evolutionary inferences about ARG dissemination.

### 2.3. Antibiotic Resistance Genes (ARGs) and Virulence Genes Prediction

ARGs and virulence genes were predicted from the P3M genome using CARD (https://card.mcmaster.ca/, accessed on 16 April 2025) (v6.0.0) [39] and VFDB (v2021) (https://www.mgc.ac.cn/VFs/main.htm, accessed on 16 April 2025) databases, respectively. For ARG prediction we applied stringent thresholds of ≥80% sequence identity and ≥80% coverage as recommended by CARD to ensure reliable detection of resistance determinants, while virulence gene identification employed slightly relaxed thresholds (≥70% identity, ≥50% coverage) following VFDB guidelines to capture functionally conserved yet evolutionarily divergent elements. These parameter selections were based on database recommendations, empirical validation studies demonstrating optimal sensitivity-specificity balance, with all analyses performed using default parameters unless otherwise specified, and full command-line documentation is available upon request to ensure complete reproducibility.

### 2.4. Phylogenetic Tree Construction

Phylogenetic analysis was performed using MEGA (v7.0.26) [40,41] to construct unrooted neighbor-joining trees of target genes and genomic islands. Coding sequences were first aligned using MUSCLE (v3.8.31) with default parameters. The Maximum Composite Likelihood (MCL) evolutionary model was selected after model testing in MEGA7. To assess node reliability, bootstrap analysis with 1000 replicates was conducted. For genomic island comparisons, phylogenetic analysis was based on core genome sequences. The resulting tree is drawn to scale, with branch lengths representing the evolutionary distances used for phylogenetic inference.

### 2.5. Sequence Alignments

Linear comparison and map generation of the genomic islands were performed and visualized using Easyfig software (v2.2.3) [42] with BLASTn (v 2.2.31+) under a 50% minimum identity threshold, where input sequences in GenBank format were annotated by Prokka and visualized as linear maps highlighting conserved syntenic blocks, boundary regions, and ARG/virulence gene clusters, followed by manual verification of domain architecture using NCBI’s CD-Search to ensure alignment accuracy.

Detailed tool specifications are provided in Appendix A.

## 3. Results

### 3.1. P3M Genomic Island Analysis

Comprehensive genomic characterization identified 16 GIs in P3M (Table 1, Figure 1), ranging from 4.4 kb (GI2) to 49.9 kb (GI6) in size and distributed throughout the chromosome, exhibiting significant structural diversity with coding gene content varying from 2 (GI14) to 69 (GI6) genes per island [43]. These GIs demonstrated notable variations in genomic location, size distribution, and gene composition, confirming their established roles in horizontal gene transfer and genome plasticity while highlighting their substantial capacity for adaptive gene acquisition and diverse functional potential.

### 3.2. Identification of Antibiotic Resistance-Associated Genomic Islands in P3M

Genomic analysis identified 218 potential ARGs in strain P3M (Appendix A: Potential ARGs on the P3M genome predicted by CARD database), with three GIs (GI7, GI13, GI16) harboring clinically significant resistance determinants (Figure 2). GI7 contained *catA* encoding chloramphenicol O-acetyltransferase [44,45] (highlighted in blue in Appendix A), while GI13 carried *hprR* associated with fluoroquinolone resistance [46] (highlighted in pink in Appendix A). GI16 possessed five ARGs (*rpoC*, *rpoB*, *tuf*, *fusA*, and *rpsL*) conferring resistance to daptomycin, rifampicin, kirromycin, fusidic acid, and aminoglycosides, respectively (highlighted in green in Appendix A), with *tuf* currently displaying resistance-conferring mutations. Notably, GI13 and GI16 contained integrases/transposases facilitating horizontal transfer, whereas GI7’s mobilization potential may involve novel mechanisms through uncharacterized ORFs. The localization of these ARGs within MGEs underscores their dissemination risk, particularly under selective pressures in aquatic environments [39].

Comparative genomic analyses of GI7, GI13, and GI16 revealed distinct evolutionary origins, with phylogenetic reconstruction (Figure 3A–C) demonstrating that GI7 showed closest affinity to *Hafnia alvei* strain 2023JQ-00054, while GI16 clustered with *Proteus penneri* strain S178-2, suggesting intergeneric horizontal transfer events. Sequence alignments (Figure 3D–F) confirmed high conservation of all ARGs across related strains, though the GI13 integrase gene appeared unique to P3M, indicating potential strain-specific mobilization mechanisms. These findings collectively demonstrate the mosaic evolutionary history of resistance GIs in P3M, with both conserved resistance determinants and strain-specific elements contributing to the dissemination.

Whole genome analysis identified 42 IS elements in P3M, predominantly from the IS200/IS605 family encoding TnpA transposase and its regulatory partner TnpB (Appendix A: IS on the P3M genome) [21,22,47]. Notably, we characterized a 30 kb genomic segment (1,609,703–1,639,229 bp) flanked by identical *tnpA*-*tnpB* genes, containing three putative ARGs (*fabG*, *pgsA*, and *satB*) with potential resistance to triclosan, daptomycin, and chloramphenicol, respectively (highlighted in yellow in Appendix A). Although *fabG* and *pgsA* currently lack resistance-conferring mutations, their unique genomic organization between functional IS elements suggests an evolutionarily active hotspot for ARG acquisition and dissemination through transposition-mediated mechanisms, representing an alternative pathway to GI-mediated ARG transfer.

Comparative genomic analysis under stringent alignment parameters (100% query coverage, >85% identity) revealed distinct conservation patterns: while *pgsA* and *satB* were universally conserved across all 13 *Proteus vulgaris* strains, *fabG* was uniquely present in P3M and 2023JQ-00005, and the flanking *tnpA*-*tnpB* genes were retained in only four strains (P3M, CCU063, 2023JQ-00005, and HH17) (Figure 4). Notably, 2023JQ-00005 maintained complete conservation of all cluster components, suggesting this strain may represent an important reservoir for IS-mediated resistance dissemination. The strain-specific distribution patterns of these MGEs highlight substantial genomic plasticity within *Proteus vulgaris* populations, likely driven by differential transposition activity across lineages. And comprehensive BLAST analysis (sequence coverage > 70%, identity > 90%) and subsequent phylogenetic reconstruction (Figure 5) revealed that the *tnpA*-*tnpB*-flanked ARG cluster in P3M exhibited closer genetic affinity to *Proteus faecis* strain 19MO01SH08 than to its conspecific *Proteus vulgaris* strain 2023JQ-00005, suggesting either horizontal acquisition from divergent *Proteus* lineages or convergent evolution under comparable selection pressures in aquatic environments.

### 3.3. Identification Virulence-Associated Genomic Islands in P3M

Genomic analysis identified over 150 virulence-associated genes in the foodborne strain P3M (Appendix A: Virulence genes on the P3M genome), with two GIs (GI12 and GI15) exhibiting significant enrichment of virulence determinants (Figure 6). GI12 contained 25 virulence genes (48.1% of its 52 total genes), including clusters for flagellar assembly and type I fimbriae synthesis, while GI15 harbored six virulence genes (40.0% of 15 genes) specifically encoding P-type fimbriae components. Both GIs carried the *xerC* integrase gene (Figure 6), a marker of horizontal transfer potential, consistent with their proposed role in disseminating motility and adhesion traits that are critical for host colonization [48,49]. The concentration of these virulence factors within MGEs suggests an efficient mechanism for pathogenicity acquisition in *Proteus vulgaris*, particularly concerning its transmission through food chains.

Phylogenetic reconstruction revealed distinct evolutionary origins for the virulence GIs, with GI12 showing closest affinity to *Proteus* sp. CD3 while GI15 clustered with *Proteus penneri* strain S178-2 (Figure 7A,B). Functional analysis demonstrated differential conservation patterns: GI12’s flagellar genes were widely conserved across species, whereas its fimbrial genes were P3M-specific; conversely, GI15’s P-type fimbrial genes showed high conservation among related species (Figure 7C,D). The unique presence of *xerC* integrase in P3M’s virulence GIs suggests strain-specific mobilization of these pathogenicity determinants, potentially contributing to its distinctive virulence profile as a foodborne pathogen.

### 3.4. Evolutionary Conservation Analysis of Genomic Islands in P3M

Comparative genomic analysis of 13 *Proteus vulgaris* strains revealed substantial variation in GI content, with GI16 emerging as the sole universally conserved GI across all strains (Figure 8A). Detailed characterization of GI16 identified 15 highly conserved genes (75% of its 20-gene content) among the 12 comparative strains, exhibiting minimal sequence variation (Figure 8B). Notably, this conserved core included all five previously identified ARGs (*rpoC*, *rpoB*, *tuf*, *fusA*, and *rpsL*), suggesting GI16 may harbor essential genetic elements maintained through purifying selection in *Proteus vulgaris* populations. The exceptional conservation of this resistance-associated island underscores its potential evolutionary significance in this bacterial species, warranting further investigation into its functional roles.

## 4. Discussion

MGEs, particularly GIs, serve as key drivers of microbial adaptation through their ability to rapidly disseminate functional genes across bacterial populations [50,51,52]. Our characterization of 16 GIs in the aquaculture-derived *Proteus vulgaris* strain P3M revealed distinct functional specialization, with two islands (GI12/GI15) enriched for virulence determinants and three (GI7/GI13/GI16) harboring ARGs. These islands exhibited remarkable structural diversity (4.4–49.9 kb) and varied genomic distributions, reflecting their dynamic evolutionary trajectories and potential for niche adaptation. Of particular significance was the identification of an IS *tnpA-tnpB*-associated ARG cluster exhibiting strain-specific conservation patterns, suggesting an alternative mobilization pathway to classical genomic island transfer. The chromosomal localization of all resistance and virulence determinants, consistent with previous reports of limited plasmid carriage in P3M [53], underscores the importance of integrative genetic elements in this strain’s adaptive arsenal. These findings collectively highlight the dual role of GIs in both enhancing *Proteus vulgaris* pathogenicity and facilitating the dissemination of resistance traits through food production chains.

Three potential mechanisms may account for these genomic arrangements: (i) Integron-mediated gene cassette capture, as observed in *Serratia*’s intI1-associated gene arrays; (ii) Tn3-like transposition via *tnpA*-*tnpB*; and (iii) Phage-derived recombination at inverted repeats flanking GI12. Notably, the unique presence of specific integrases in resistance-associated GI13 and virulence-associated GI12/GI15 implies P3M has evolved distinctive genetic transfer strategies compared to other *Proteus* strains, providing novel insights into the evolutionary dynamics of genomic islands in environmental pathogens. These findings significantly expand our understanding of how genomic plasticity contributes to bacterial adaptation in aquaculture systems.

From a One Health standpoint, the persistence of these GIs in foodborne *Proteus vulgaris* underscores important concerns about potential resistance gene dissemination to human-associated pathogens [54]. While our bioinformatic analyses reveal compelling patterns of genomic island conservation and diversification, the study has inherent limitations that warrant acknowledgment. Future investigations should prioritize experimental validation through functional genomics approaches, including targeted gene knockout studies and comprehensive characterization of resistance and virulence gene expression profiles under relevant environmental conditions. Such efforts will be crucial for elucidating the precise mechanisms governing antibiotic resistance development and pathogenicity in this emerging foodborne pathogen, ultimately informing more effective surveillance and control strategies at the human–animal-environment interface.

## 5. Conclusions

This study provides important genomic insights into the multidrug-resistant foodborne pathogen *Proteus vulgaris* strain P3M, characterizing 16 GIs with specialized functional profiles, including virulence-enriched GI12/GI15 and resistance-associated GI7/GI13/GI16. The universal conservation of GI16 across *Proteus vulgaris* strains contrasts with strain-specific features like GI13’s unique integrase, suggesting complex adaptation strategies in aquaculture environments. Although our bioinformatic analyses reveal compelling evidence for horizontal gene transfer potential, the study’s limitations—particularly the need for functional validation of predicted resistance/virulence genes and experimental confirmation of genomic island mobility—highlight important directions for future research. Moving forward, integrating transcriptomic profiling, conjugation assays, and expanded environmental surveillance will be essential to fully elucidate the transmission dynamics of these genetic elements and develop effective control measures at the aquaculture–human interface, ultimately contributing to more robust antimicrobial resistance containment strategies within the One Health framework.

## Figures and Tables

**Figure 1 biology-14-00858-f001:**
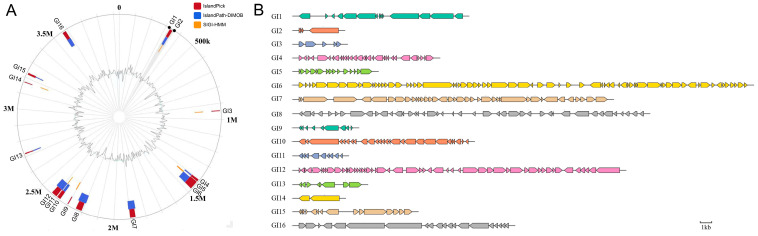
Genomic island distribution and gene content in *Proteus vulgaris* strain P3M. (**A**) Circular genome map showing locations of 16 predicted GIs (GI1–GI16) with IslandViewer4. Colors represent different prediction methods. (**B**) Bar plot of coding gene counts per GI (range: 2–69 genes). Scale bar: 1 kb.

**Figure 2 biology-14-00858-f002:**
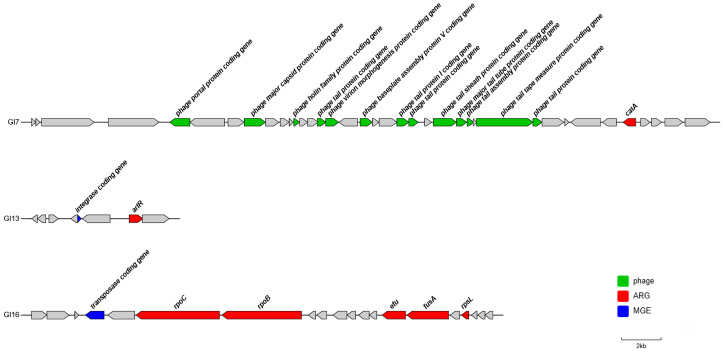
Distribution of ARGs on GI7, GI13, and GI16.

**Figure 3 biology-14-00858-f003:**
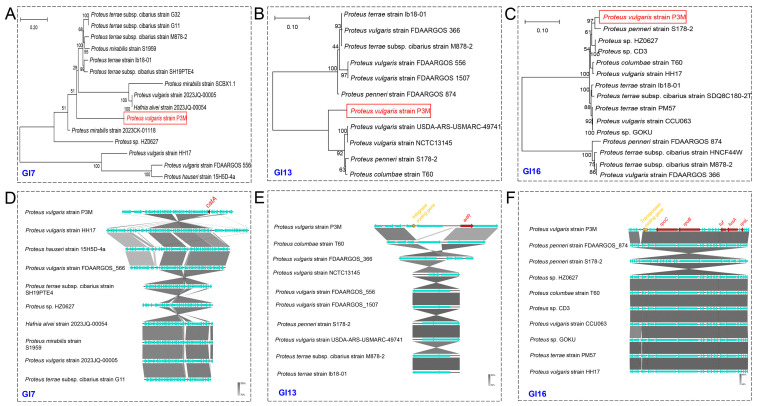
Phylogenetic analysis of resistance-associated GIs. (**A**–**C**) Neighbor-joining trees of GI7/GI13/GI16 core sequences. Bootstrap values > 70% shown. Scale bar: 0.05 substitutions per site. (**D**–**F**) BLASTn alignments (Easyfig) with closely related species.

**Figure 4 biology-14-00858-f004:**
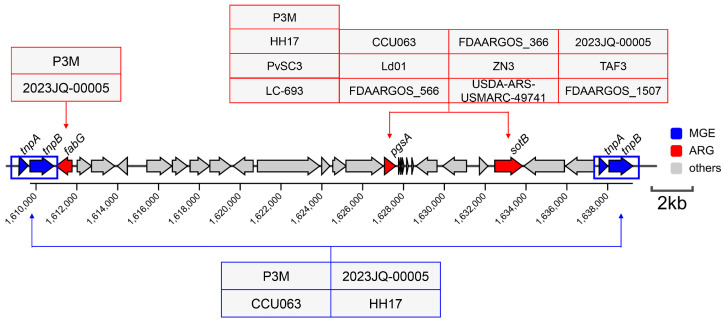
Conservation analysis of *tnpA-tnpB*-linked ARG cluster.

**Figure 5 biology-14-00858-f005:**
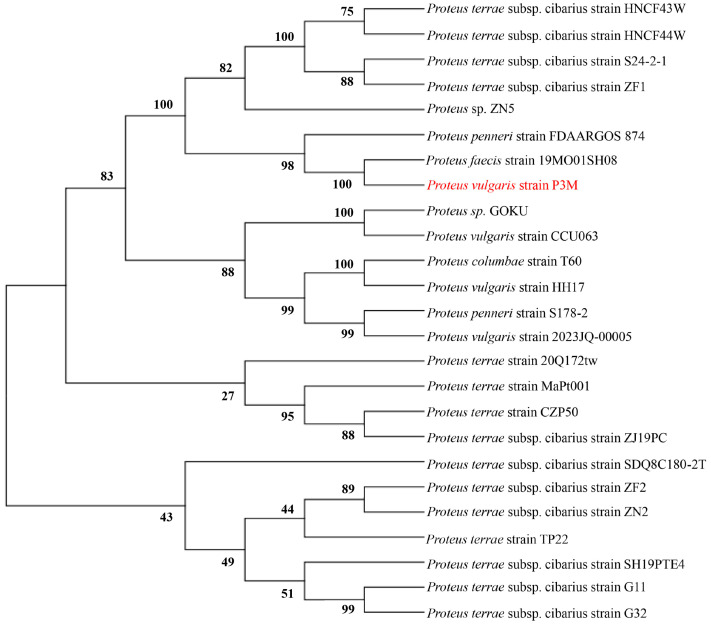
Phylogenetic analysis of the *tnpA*-*tnpB*-linked ARG cluster. Maximum-likelihood tree based on concatenated tnpA-tnpB sequences (1000 bootstrap replicates).

**Figure 6 biology-14-00858-f006:**
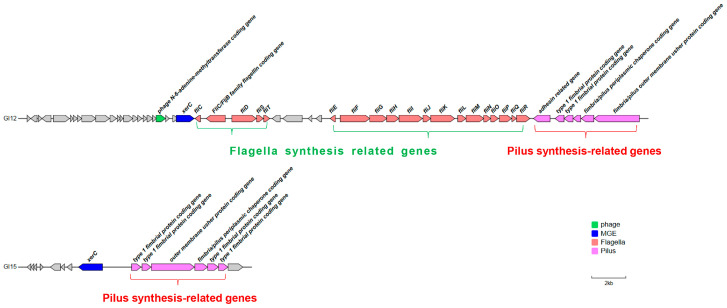
Distribution of virulence genes on GI12 and GI15.

**Figure 7 biology-14-00858-f007:**
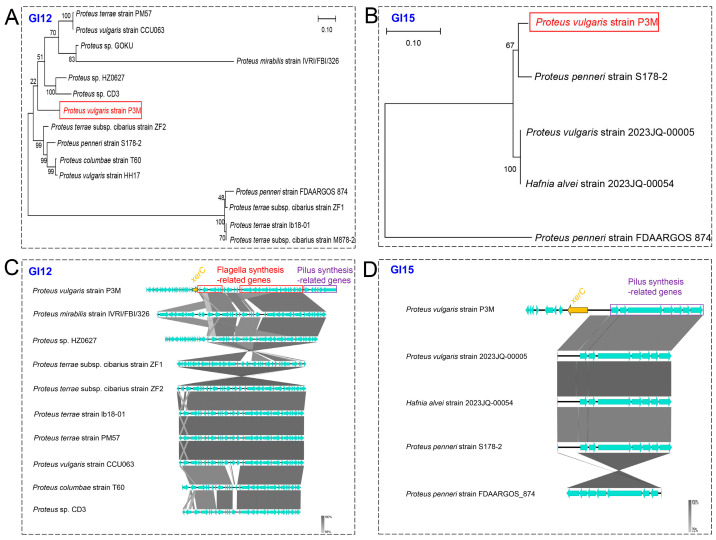
Phylogenetic analysis and linear alignment of GI12 and GI15. Phylogenetic analysis of virulence-associated GIs. (**A**,**B**) Neighbor-joining trees of GI12/GI15 core sequences. (**C**,**D**) BLASTn alignments (Easyfig) with closely related species.

**Figure 8 biology-14-00858-f008:**
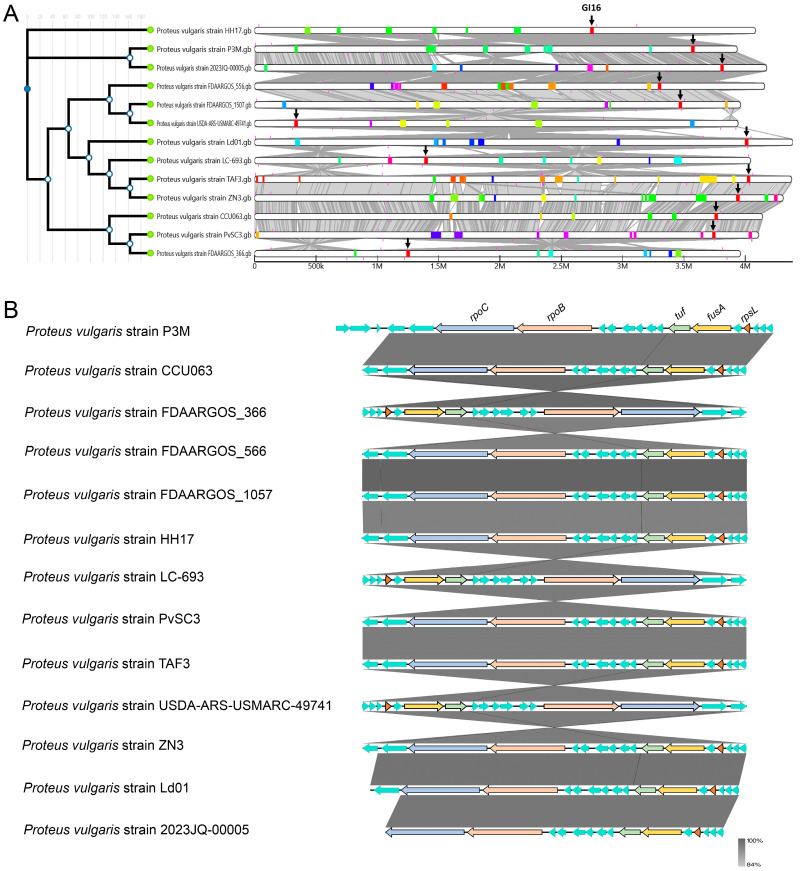
(**A**) Alignment of genomic islands in whole genome-sequenced *Proteus vulgaris* strains. (**B**) Linear alignment of GI16 among whole-genome sequenced *Proteus vulgaris* strains.

**Table 1 biology-14-00858-t001:** Predicted genomic islands from the P3M genome (obtained using IslandViewer 4).

Genomic Island (GI)	Start	End	Size
GI1	330,090	348,336	18,246
GI2	339,053	343,460	4407
GI3	943,652	948,353	4701
GI4	1,401,531	1,415,713	14,182
GI5	1,418,845	1,426,992	8147
GI6	1,423,836	1,473,830	49,994
GI7	1,865,071	1,899,237	34,166
GI8	2,206,196	2,244,624	38,428
GI9	2,299,849	2,305,598	5749
GI10	2,358,993	2,377,861	18,868
GI11	2,378,798	2,383,241	4443
GI12	2,388,800	2,424,554	35,754
GI13	2,714,880	2,721,564	6684
GI14	3,170,960	3,175,456	4496
GI15	3,221,960	3,234,561	12,601
GI16	3,561,510	3,584,877	23,367

## Data Availability

Data supporting the findings of the present study are included in the article and Appendix A. The complete genomic information of *Proteus vulgaris* strains can be accessed through the database (https://www.ncbi.nlm.nih.gov/nucleotide/, accessed on 11 April 2025).

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
