# Peer review of "Unveiling Genomic Islands Hosting Antibiotic Resistance Genes and Virulence Genes in Foodborne Multidrug-Resistant Patho-Genic Proteus vulgaris"

_biology, 2025, doi:10.3390/biology14070858_

Round 1

Reviewer 1 Report

Comments and Suggestions for Authors

The manuscript titled "Unveiling genomic islands hosting antibiotic resistance genes and virulence genes in foodborne multidrug-resistant pathogenic Proteus vulgaris" addresses an important and timely topic in microbial genomics and antimicrobial resistance. The focus on Proteus vulgaris and the identification of genomic islands that co-harbor ARGs and virulence genes represent relevant contributions to the field.

However, the manuscript requires substantial revisions before it can be considered for publication. The abstract and introduction should emphasize the study's novelty more effectively and clearly state the research question. While the methods are generally adequate, they should be more transparent, particularly concerning the bioinformatics pipelines and gene identification tools. The results would benefit from a clearer structure, and the discussion should include comparative analyses with related Enterobacterales to strengthen the interpretation of the findings. Figures and tables are informative, although some need reformatting for clarity and consistency. Finally, the quality of the English should be thoroughly reviewed to ensure readability.

Specific comments by section are provided in the attached file. These suggestions aim to improve the scientific communication and impact of the manuscript.

Reviewer Comments – Manuscript ID: biology-3700751
Title: Unveiling genomic islands hosting antibiotic resistance genes and virulence genes
in foodborne multidrug-resistant pathogenic Proteus vulgaris.
Authors: Hongyang Zhang , Tao Wu , Haihua Ruan *

Abstract

The abstract presents the topic clearly and introduces Proteus vulgaris as a multidrug-
resistant foodborne pathogen. However, to enhance scientific clarity and impact, I
recommend the following: 1. Include more specific data from the results to highlight key
findings, such as the number or types of resistance genes detected, the prevalence of
virulence factors, or notable genomic islands. 2. Briefly clarify the methodology—e.g.,
whole genome sequencing (WGS) and GI identification tools—so the reader understands
how the conclusions were reached. 3. Simplify sentence structure to increase clarity and
avoid ambiguity. For example, breaking up long sentences or restructuring passive
sentences would improve readability. 4. Strengthen the conclusion by explicitly linking
the results to broader public health implications, such as the role of MDR P. vulgaris in
food safety or the relevance to the One Health framework.

Introduction

The introduction outlines the relevance of Proteus vulgaris as an opportunistic foodborne
pathogen and highlights the role of genomic islands in the co-localization of antibiotic
resistance and virulence genes. However, to strengthen the scientific rationale of your
study, I suggest the following improvements:

1. Expand the ecological and epidemiological context of P. vulgaris in aquaculture
environments, particularly in shrimp or similar foodborne sources. This would help
justify the choice of organism and sample origin more robustly.

2. Update and expand references to include more recent studies (from the last 3–5
years) that address the emergence of multidrug-resistant Proteus spp. in food sources and
their genomic plasticity. This would reinforce the originality and timeliness of your work.

3. Consider briefly introducing the One Health implications of your findings to
connect the introduction with the broader public health context discussed later.

Research Design

The research design is appropriate for the study's scope. However, additional justification
for selecting the genomic comparison methods and the strain P3M should be included to
enhance the rationale.

Materials and Methods

The Materials and Methods section explains how to characterize the genome of Proteus
vulgaris, but there are a few elements that need to be clarified or expanded to ensure full
reproducibility and scientific rigor.

Strain origin and identification:

Please include detailed information on the isolation and identification of the P. vulgaris
P3M strain isolated from Penaeus vannamei. Specify the date and source sample was
collected, culture media used, and confirmatory identification methods (e.g., MALDI-
TOF, 16S rRNA sequencing, or API test).

Sequencing and genome assembly:

The manuscript should clearly state the sequencing platform used, the library preparation
protocol, read quality filtering steps, and genome assembly software. Report basic
assembly metrics (e.g., number of contigs, N50, genome length, and completeness using
tools like QUAST or BUSCO).

Bioinformatics tools and thresholds:

While tools such as IslandViewer 4, IslandCompare, CARD-RGI, and Easyfig are
appropriate, the manuscript should specify: a) the version number of each tool and
associated databases (e.g., CARD, VFDB), b) the identity and coverage thresholds used
for ARG and virulence gene prediction.

Overly procedural descriptions:

Avoid step-by-step instructions for web interfaces (e.g., “click on Analyze, then
Upload”), which reduce the scientific tone. Instead, focus on describing analytical logic
and inputs/outputs.

Genome dataset justification:

Clarify why the 12 comparative P. vulgaris genomes were selected. Were they chosen for
completeness, clinical relevance, host origin, or availability?

Phylogenetic and comparative analyses:

While MEGA7 is referenced for tree construction, the following should be included:
alignment method (e.g., MUSCLE), evolutionary model (e.g., Maximum Composite
Likelihood), whether bootstrap support was used (and how many replicates), and what
sequences were included (e.g., core genome vs. selected markers).

Reproducibility enhancement:

It is possible to include a brief table summarizing all bioinformatics tools, versions,
databases, and analysis platforms with their URLs. Indicate whether tools were run online
or locally (e.g., command-line vs. web-based versions).

Results

The Results section addresses important findings regarding the co-occurrence of
antibiotic resistance and virulence genes in Proteus vulgaris; however, several aspects
could be improved for better clarity and reader engagement:

1. Highlight specific findings in the main text from Tables S1–S3. For instance,
mention the most prevalent resistance genes or key virulence factors identified.

2. Summarize the functional categories of resistance and virulence genes (e.g.,
efflux pumps, toxins, adhesion proteins) and discuss their implications in pathogenesis or
treatment failure.

3. Clarify the genomic context of resistance and virulence gene localization within
genomic islands. Consider adding a figure showing representative genomic island
structures.

4. Add interpretive descriptions to each table either in the main text or table
captions, so readers can understand the relevance of the data without switching back and
forth.

Discussion

The discussion interprets the results broadly but needs deeper comparative context,
particularly with related Enterobacterales (E. coli, Salmonella, Serratia). Discuss
potential mechanisms for the co-localization of virulence and resistance genes (e.g.,
integrons, transposons, or phage-mediated transfer), and how this might contribute to the
pathogen's adaptation in aquaculture or One Health environments.

Conclusions

The conclusions are valid but can be made more impactful by directly linking the
findings to One Health concerns. Future directions are briefly mentioned; they could be
expanded to reinforce the study’s relevance. A revised paragraph is recommended to
summarize the implications more clearly.

Supplementary Tables (1s–3s)

The supplementary tables provide detailed information on genomic islands, resistance
genes, and virulence factors identified in Proteus vulgaris. However, we recommend
improving the clarity and utility of these tables by adding legends that describe the
classification criteria, grouping genes by function or antimicrobial class, and indicating
whether the genes are plasmid- or chromosome-borne when known. Including database
references or detection tools for gene identification would also enhance transparency and
reproducibility. Formatting adjustments (e.g., bold headers, row numbers) would improve
readability for the reader.

References

The references used are pertinent and validate the genomic, epidemiological, and One
Health aspects of the study. However, the list could be improved by incorporating more
recent reviews or systematic analyses published in the last 3–5 years, particularly those
focusing on resistance gene dissemination and genomic islands in foodborne pathogens.
It would also be beneficial to cite the bioinformatics tools and databases employed for
gene identification and annotation.

Comments on the Quality of English Language

While the manuscript is generally understandable and well organized, there are issues with word choice, grammar, and clarity of expression in the English text. Some expressions are repetitive or ambiguous, and several technical sections would benefit from clearer phrasing to make them more readable and polished for an academic audience. Improving the language would significantly enhance the article's readability.

Author Response

Response to Reviewer 1 Comments

1. Summary

2. Questions for General Evaluation

Reviewer’s Evaluation

Response and Revisions

Does the introduction provide sufficient background and include all relevant references?

Can be improved

We sincerely appreciate your valuable comments and constructive suggestions. In response to these insightful recommendations, we have meticulously revised the manuscript, addressing each point in detail to enhance the overall quality of the work and ensure it meets the rigorous publication standards of your esteemed journal.

Are all the cited references relevant to the research?

Can be improved

Is the research design appropriate?

Can be improved

Are the methods adequately described?

Can be improved

Are the results clearly presented?

Can be improved

Are the conclusions supported by the results?

Can be improved

3. Point-by-point response to Comments and Suggestions for Authors

Comments 1:

Abstract

The abstract presents the topic clearly and introduces Proteus vulgaris as a multidrug-resistant foodborne pathogen. However, to enhance scientific clarity and impact, I recommend the following: 1. Include more specific data from the results to highlight key findings, such as the number or types of resistance genes detected, the prevalence of virulence factors, or notable genomic islands. 2. Briefly clarify the methodology—e.g., whole genome sequencing (WGS) and GI identification tools—so the reader understands how the conclusions were reached. 3. Simplify sentence structure to increase clarity and avoid ambiguity. For example, breaking up long sentences or restructuring passive sentences would improve readability. 4. Strengthen the conclusion by explicitly linking the results to broader public health implications, such as the role of MDR P. vulgaris in food safety or the relevance to the One Health framework.

Response 1: We appreciate your valuable suggestions and have carefully revised the Abstract section by incorporating more specific data presentation, briefly clarifying methodological details, simplifying sentence structures for better readability, and emphasizing the significance of our research findings.

Comments 2:

Introduction

The introduction outlines the relevance of Proteus vulgaris as an opportunistic foodborne pathogen and highlights the role of genomic islands in the co-localization of antibiotic resistance and virulence genes. However, to strengthen the scientific rationale of your study, I suggest the following improvements:

1. Expand the ecological and epidemiological context of P. vulgaris in aquaculture environments, particularly in shrimp or similar foodborne sources. This would help justify the choice of organism and sample origin more robustly.

Response 2: Thank you for your valuable suggestion. We agree with you and have expanded the ecological and epidemiological context of P. vulgaris in aquaculture environments within the Introduction section to provide a more robust scientific rationale for our sample selection.

Comments 3:

Introduction

2. Update and expand references to include more recent studies (from the last 3–5 years) that address the emergence of multidrug-resistant Proteus spp. in food sources and their genomic plasticity. This would reinforce the originality and timeliness of your work.

Response 3: Thank you for your valuable suggestion. We have carefully updated the reference list with the most recent and relevant references to strengthen the scientific foundation and highlight the novelty and contemporary significance of our research.

Comments 4:

Introduction

3. Consider briefly introducing the One Health implications of your findings to connect the introduction with the broader public health context discussed later.

Response 4: We appreciate this suggestion. We have revised the Introduction's concluding paragraph to explicitly highlight the One Health implications of our findings, emphasizing how P. vulgaris strain P3M serves as a potential reservoir for ARG dissemination across aquatic ecosystems and human populations, thereby strengthening the connection to broader public health impacts.

Comments 5:

Research Design

The research design is appropriate for the study's scope. However, additional justification for selecting the genomic comparison methods and the strain P3M should be included to enhance the rationale.

Response 5: Thank you for pointing this out. We have strengthened the justification for selecting strain P3M in the Introduction's concluding paragraph. We also clarified the rationale for selecting our genomic comparison methods in Section 2.2.

Comments 6:

Materials and Methods

The Materials and Methods section explains how to characterize the genome of Proteus vulgaris, but there are a few elements that need to be clarified or expanded to ensure full reproducibility and scientific rigor.

Response 6: We appreciate your valuable comments regarding methodological clarity. In this study, all 13 Proteus vulgaris strains (including P3M) analyzed were complete genome sequences obtained from NCBI, with sequencing and assembly methodologies rigorously documented in their original submissions by various research institutions. To ensure full reproducibility, we have: (1) provided all NCBI accession numbers (e.g., CP060211 for P3M) in Section 2.1, allowing direct access to original sequencing metadata; and (2) added explicit clarification in Methods Section 2.1 that these are complete, publicly available genomes with established quality metrics.

Comments 7:

Strain origin and identification:

Please include detailed information on the isolation and identification of the P. vulgaris P3M strain isolated from Penaeus vannamei. Specify the date and source sample was collected, culture media used, and confirmatory identification methods (e.g., MALDI-TOF, 16S rRNA sequencing, or API test).

Response 7: Thank you for this constructive suggestion. The complete isolation and identification protocols for P. vulgaris P3M, including sample collection details, culture conditions, and confirmatory identification methods were thoroughly documented in our previous publication (Reference 28 in the revised version). As this strain's characterization has been formally established and peer-reviewed, we referenced these methods rather than repeating them to maintain conciseness while ensuring full methodological transparency. The strain's genome (CP060211) and associated metadata remain publicly available in NCBI for independent verification.

Comments 8:

Sequencing and genome assembly:

The manuscript should clearly state the sequencing platform used, the library preparation protocol, read quality filtering steps, and genome assembly software. Report basic assembly metrics (e.g., number of contigs, N50, genome length, and completeness using tools like QUAST or BUSCO).

Response 8: Thank you for raising this methodological consideration. The complete genome sequencing of P. vulgaris P3M and all other analyzed strains were conducted in previous studies, with comprehensive technical details including sequencing platforms, library preparation protocols, assembly pipelines, and quality metrics already documented in their respective accessions provided in Section 2.1. Since our study focused on comparative analysis of these established complete genomes rather than conducting new sequencing, we referenced these verified resources to maintain conciseness while ensuring full methodological traceability. All assembly statistics remain publicly accessible through the provided NCBI entries for independent verification.

Comments 9:

Bioinformatics tools and thresholds:

While tools such as IslandViewer 4, IslandCompare, CARD-RGI, and Easyfig are appropriate, the manuscript should specify: a) the version number of each tool and associated databases (e.g., CARD, VFDB), b) the identity and coverage thresholds used for ARG and virulence gene prediction.

Response 9: We appreciate your suggestion. We have now specified the version numbers of all bioinformatics tools used and clearly stated the coverage thresholds applied for ARG and virulence gene predictions in the revised Methods section.

Comments 10:

Overly procedural descriptions:

Avoid step-by-step instructions for web interfaces (e.g., “click on Analyze, then Upload”), which reduce the scientific tone. Instead, focus on describing analytical logic and inputs/outputs.

Response 10: We appreciate your constructive feedback. We have revised the Methods section to focus on the scientific rationale and key parameters of our analyses.

Comments 11:

Genome dataset justification:

Clarify why the 12 comparative P. vulgaris genomes were selected. Were they chosen for completeness, clinical relevance, host origin, or availability?

Response 11: We appreciate your inquiry regarding strain selection. The 12 additional Proteus vulgaris genomes were included in our analysis because they represent all currently available complete genome sequences of this species in public databases. This comprehensive approach enables robust comparative genomics and enhances the statistical power of our evolutionary analyses. We have clarified this rationale in Section 2.1.

Comments 12:

Phylogenetic and comparative analyses:

While MEGA7 is referenced for tree construction, the following should be included: alignment method (e.g., MUSCLE), evolutionary model (e.g., Maximum Composite Likelihood), whether bootstrap support was used (and how many replicates), and what sequences were included (e.g., core genome vs. selected markers).

Response 12: Thank you for your constructive suggestion. We have thoroughly revised Section 2.4 to enhance methodological clarity and scientific rigor, specifically addressing each of your valuable recommendations regarding phylogenetic analysis.

Comments 13:

Reproducibility enhancement:

It is possible to include a brief table summarizing all bioinformatics tools, versions, databases, and analysis platforms with their URLs. Indicate whether tools were run online or locally (e.g., command-line vs. web-based versions).

Response 13: Thank you for this constructive suggestion. As requested, we have added Supplementary Table S4 summarizing detailed information of all bioinformatics tools used in this study. (Please refer to Attachment Table S4)

Comments 14:

Results

The Results section addresses important findings regarding the co-occurrence of antibiotic resistance and virulence genes in Proteus vulgaris; however, several aspects could be improved for better clarity and reader engagement:

1. Highlight specific findings in the main text from Tables S1–S3. For instance, mention the most prevalent resistance genes or key virulence factors identified.

Response 14: We appreciate your suggestions. In this study, Tables S1-S3 are served as comprehensive repositories supporting the findings detailed in our main text. Specifically: (1) Table S1 systematically catalogs all 218 putative ARGs identified in P3M, with relevant resistance genes (catA, hprR, rpoC, rpoB, tuf, fusA, rpsL) highlighted in Results section 3.2; (2) Table S2 documents virulence factors, with particular emphasis in the text on the flagellar/fimbrial gene clusters in GI12/GI15 (Results 3.3); (3) Table S3’s IS element analysis underpins the mobile genetic element discussion in Results 3.4. This selective presentation ensures the main text maintains focus on genomic islands while the supplementary tables provide full methodological transparency.

Comments 15:

Results

2. Summarize the functional categories of resistance and virulence genes (e.g., efflux pumps, toxins, adhesion proteins) and discuss their implications in pathogenesis or treatment failure.

Response 15: We appreciate your suggestion regarding functional characterization of resistance and virulence genes. Our study primarily focuses on elucidating the genomic island-mediated transfer of these genetic determinants, as detailed in Results sections 3.2 (for resistance islands GI7/13/16) and 3.3 (for virulence islands GI12/15). While we have described the key functional categories of these genes (e.g., antibiotic modification enzymes, adhesion factors), we have intentionally maintained our emphasis on their horizontal transfer potential rather than their specific pathogenic mechanisms, to preserve the study's central focus on genome plasticity in P. vulgaris. The current level of functional annotation strikes an appropriate balance between contextualizing the biological significance of these genes and maintaining the paper's core genomic island focus.

Comments 16:

Results

3. Clarify the genomic context of resistance and virulence gene localization within genomic islands. Consider adding a figure showing representative genomic island structures.

Response 16: We appreciate your suggestion and wish to clarify that our integrated presentation strategy was designed to comprehensively characterize genomic island architecture through complementary data representations: Table 1 provides precise genomic coordinates and physical dimensions of all identified islands, while Figures 2 and 6 respectively illustrate the spatial organization of antibiotic resistance genes (GI7/GI13/GI16) and virulence factors (GI12/GI15), enabling simultaneous assessment of structural features (size/location via Table 1) and functional elements (gene clusters via figures) - an approach that optimally balances quantitative positional data with functional annotation while maintaining graphical clarity and scientific rigor.

Comments 17:

Results

4. Add interpretive descriptions to each table either in the main text or table captions, so readers can understand the relevance of the data without switching back and forth.

Response 17: Thank you for your constructive suggestion. We have detailed description of the supplementary Tables mentioned in the main text, ensuring to enhance the correlation and logic among the data.

Comments 18:

Discussion

The discussion interprets the results broadly but needs deeper comparative context, particularly with related Enterobacterales (E. coli, Salmonella, Serratia). Discuss potential mechanisms for the co-localization of virulence and resistance genes (e.g., integrons, transposons, or phage-mediated transfer), and how this might contribute to the pathogen's adaptation in aquaculture or One Health environments.

Response 18: We appreciate your valuable suggestions for improving the Discussion section. In response, we have thoroughly revised this section, provided deeper comparative analysis and strengthened the discussion of our findings' implications for aquaculture environments and One Health contexts. We believe these modifications have significantly enhanced the section's scientific depth and translational relevance while maintaining clear focus on our core findings.

Comments 19:

Conclusions

The conclusions are valid but can be made more impactful by directly linking the findings to One Health concerns. Future directions are briefly mentioned; they could be expanded to reinforce the study’s relevance. A revised paragraph is recommended to summarize the implications more clearly.

Response 19: We appreciate your constructive suggestions to strengthen the impact of our conclusions. We have revised the Conclusion section to explicitly connect our genomic findings to One Health concerns, and expand future research directions to highlight their potential for developing targeted surveillance strategies.

Comments 20:

Supplementary Tables (1s–3s)

The supplementary tables provide detailed information on genomic islands, resistance genes, and virulence factors identified in Proteus vulgaris. However, we recommend improving the clarity and utility of these tables by adding legends that describe the classification criteria, grouping genes by function or antimicrobial class, and indicating whether the genes are plasmid- or chromosome-borne when known. Including database references or detection tools for gene identification would also enhance transparency and reproducibility. Formatting adjustments (e.g., bold headers, row numbers) would improve readability for the reader.

Response 20: We sincerely appreciate your valuable suggestion. The supplementary tables (S1-S3) in this study provide comprehensive annotations of the P3M genome, including: (1) antibiotic resistance genes with their respective mechanisms and target antibiotics, (2) insertion sequences with family classifications and genomic locations, and (3) virulence factors with functional characterizations. All study genes were confirmed as chromosomally encoded, with the two small plasmids identified in P3M (Reference 50 in the revised version) containing no relevant resistance or virulence genes. We have provided supplementary explanations on this point in the Discussion section of the revised manuscript.

Comments 21:

References

The references used are pertinent and validate the genomic, epidemiological, and One Health aspects of the study. However, the list could be improved by incorporating more recent reviews or systematic analyses published in the last 3–5 years, particularly those focusing on resistance gene dissemination and genomic islands in foodborne pathogens. It would also be beneficial to cite the bioinformatics tools and databases employed for gene identification and annotation.

Response 21: We sincerely your constructive suggestions for strengthening our reference list. In response, we have systematically updated the references to enhance the literature foundation while maintaining focus on our core findings.

4. Response to Comments on the Quality of English Language

Point 1: While the manuscript is generally understandable and well organized, there are issues with word choice, grammar, and clarity of expression in the English text. Some expressions are repetitive or ambiguous, and several technical sections would benefit from clearer phrasing to make them more readable and polished for an academic audience. Improving the language would significantly enhance the article's readability.

Response 1: We sincerely appreciate your insightful suggestions. In response, we have carefully refined the manuscript's language to enhance scientific precision, improve clarity, and strengthen overall readability while maintaining rigorous academic standards.

5. Additional clarifications

We sincerely appreciate your thorough review and constructive suggestions, which have greatly enhanced our manuscript. In response to your comments, we have carefully addressed each point, conducted a comprehensive language edit for clarity and precision, and verified all technical content for accuracy. To facilitate your review, we are submitting both a version with tracked changes highlighting all modifications and a clean final version. We are truly grateful for the time and expertise you have dedicated to evaluating our work and would be happy to make any additional revisions if needed.

Reviewer 2 Report

Comments and Suggestions for Authors

This manuscript presents a bioinformatics analysis of genomic islands in Proteus vulgaris strain P3M. While antimicrobial resistance in foodborne pathogens is an important research area, several fundamental issues limit the scientific contribution and publication readiness of this work.

Major Issues:

1. The study lacks a clearly defined research hypothesis and represents a descriptive bioinformatics analysis without experimental validation. The findings that a multidrug-resistant pathogen harbors genomic islands with resistance and virulence genes are predictable and require experimental confirmation to advance scientific understanding.

2. The Materials and Methods section contains excessive software interface instructions inappropriate for scientific literature. Please focus on analytical approaches, parameter justification, and reproducible methodology. Citations supporting the selection of coverage and identity thresholds for sequence comparisons are required.

3. Persistent misuse of "Proteus vulgaris species" throughout the manuscript. Correct usage is "Proteus vulgaris strains" when referring to different isolates of the same species. 

4. The manuscript lacks phenotypic antimicrobial susceptibility data to validate computational resistance predictions. Without experimental confirmation, conclusions regarding resistance mechanisms remain speculative.

5. The Discussion section inadequately analyzes the results, largely reiterating findings without critical interpretation or contextualization within existing literature on Proteus genomics and horizontal gene transfer.

Minor Issues:

All figures require substantial improvement in resolution and clarity to meet publication standards (minimum 300 dpi).

Comments on the Quality of English Language

The manuscript requires comprehensive English language editing for clarity and scientific precision.

Author Response

Response to Reviewer 2 Comments

1. Summary

2. Questions for General Evaluation

Reviewer’s Evaluation

Response and Revisions

Does the introduction provide sufficient background and include all relevant references?

Can be improved

We sincerely appreciate your valuable comments and constructive suggestions. In response to these insightful recommendations, we have meticulously revised the manuscript, addressing each point in detail to enhance the overall quality of the work and ensure it meets the rigorous publication standards of your esteemed journal.

Are all the cited references relevant to the research?

Can be improved

Is the research design appropriate?

Must be improved

Are the methods adequately described?

Must be improved

Are the results clearly presented?

Can be improved

Are the conclusions supported by the results?

Must be improved

3. Point-by-point response to Comments and Suggestions for Authors

Comments 1:

1. The study lacks a clearly defined research hypothesis and represents a descriptive bioinformatics analysis without experimental validation. The findings that a multidrug-resistant pathogen harbors genomic islands with resistance and virulence genes are predictable and require experimental confirmation to advance scientific understanding.

Response 1: We sincerely appreciate your insightful suggestion regarding experimental validation of the predicted genomic islands. We fully concur that functional verification through molecular biology approaches represents a crucial next step, and we have accordingly incorporated this important perspective into our future research plans. Specifically, we intend to employ gene knockout and site-directed mutagenesis techniques to experimentally validate the genomic islands identified in this bioinformatic study. While these validation experiments will require substantial time and resources, the predictive findings from our current work will serve as a valuable foundation to guide these future investigations.

Comments 2:

2. The Materials and Methods section contains excessive software interface instructions inappropriate for scientific literature. Please focus on analytical approaches, parameter justification, and reproducible methodology. Citations supporting the selection of coverage and identity thresholds for sequence comparisons are required.

Response 2: We appreciate your constructive feedback. We have revised the Methods section to focus on the scientific rationale and key parameters of our analyses, avoiding step-by-step instructions for web interfaces.

Comments 3:

3. Persistent misuse of "Proteus vulgaris species" throughout the manuscript. Correct usage is "Proteus vulgaris strains" when referring to different isolates of the same species.

Response 3: We appreciate your careful attention to terminology precision. We have systematically corrected all instances where “Proteus vulgaris species” was inappropriately used, replacing them with the accurate designation “Proteus vulgaris strains” throughout the manuscript.

Comments 4:

4. The manuscript lacks phenotypic antimicrobial susceptibility data to validate computational resistance predictions. Without experimental confirmation, conclusions regarding resistance mechanisms remain speculative.

Response 4: We sincerely appreciate your valuable suggestion regarding experimental validation. While this bioinformatics study primarily focuses on the preliminary identification of genomic islands in Proteus vulgaris strain P3M, we fully acknowledge the importance of experimental confirmation for resistance prediction accuracy and have therefore incorporated mechanistic validation of ARG-bearing genomic islands as a key objective in our future research plans, which will employ both molecular and phenotypic approaches to elucidate the specific resistance mechanisms involved.

Comments 5:

5. The Discussion section inadequately analyzes the results, largely reiterating findings without critical interpretation or contextualization within existing literature on Proteus genomics and horizontal gene transfer.

Response 5: We appreciate your valuable suggestions for improving the Discussion section. In response, we have thoroughly revised this section, strengthening the discussion of our findings' implications for horizontal gene transfer and One Health contexts. We believe these modifications have significantly enhanced the section's scientific depth and translational relevance while maintaining clear focus on our core findings.

Comments 6:

All figures require substantial improvement in resolution and clarity to meet publication standards (minimum 300 dpi).

Response 6: Thank you for your valuable suggestions. In response to your comments, we have comprehensively revised all Figures in the manuscript. And high-resolution versions of these Figures have been prepared and uploaded to the online submission system to ensure they meet the journal's publication standards for clarity and quality.

4. Response to Comments on the Quality of English Language

Point 1: The manuscript requires comprehensive English language editing for clarity and scientific precision.

Response 1: We sincerely appreciate your insightful suggestions. In response, we have carefully refined the manuscript's language to enhance scientific precision, improve clarity, and strengthen overall readability while maintaining rigorous academic standards.

5. Additional clarifications

We sincerely appreciate your thorough review and constructive suggestions, which have greatly enhanced our manuscript. In response to your comments, we have carefully addressed each point, conducted a comprehensive language edit for clarity and precision, and verified all technical content for accuracy. To facilitate your review, we are submitting both a version with tracked changes highlighting all modifications and a clean final version. We are truly grateful for the time and expertise you have dedicated to evaluating our work and would be happy to make any additional revisions if needed.

Reviewer 3 Report

Comments and Suggestions for Authors

The manuscript titled ‘ Unveiling genomic islands hosting antibiotic resistance genes and virulence genes in foodborne multidrug-resistant pathogenic Proteus vulgaris’ is to most extent well-written and clearly articulated.

Introduction sets the context and significance of the research, methodology used is appropriate and results interpretation is valid. However, the following points need to be addressed.

  1. Language used in material and methods doesn't look scientific and needs to be rewritten
  2. Figures are blurry, with low resolution and unreadable. Need to incorporate good quality images in order to interpret the data and validate the conclusions made.
  3. It would be nice if the relevant genomic island locations are incorporated in the table S1, so that all relevant data is at one place to access.
  4. Discussion part needs to be elaborated to add weightage to the statement 'This finding contributes to our understanding of the mechanisms behind the dissemination of pathogenicity and antibiotic resistance traits in foodborne Proteus vulgaris strains, particularly from the perspective of gene transfer' written in the introduction. 

Comments on the Quality of English Language

Materials and methods section is written in non-scientific manner and needs improvement.

Author Response

Response to Reviewer 3 Comments

1. Summary

2. Questions for General Evaluation

Reviewer’s Evaluation

Response and Revisions

Does the introduction provide sufficient background and include all relevant references?

Yes

We sincerely appreciate your valuable comments and constructive suggestions. In response to these insightful recommendations, we have meticulously revised the manuscript, addressing each point in detail to enhance the overall quality of the work and ensure it meets the rigorous publication standards of your esteemed journal.

Are all the cited references relevant to the research?

Yes

Is the research design appropriate?

Yes

Are the methods adequately described?

Yes

Are the results clearly presented?

Must be improved

Are the conclusions supported by the results?

Can be improved

3. Point-by-point response to Comments and Suggestions for Authors

Comments 1:

1.         Language used in material and methods doesn't look scientific and needs to be rewritten.

Response 1: We appreciate your constructive feedback. We have revised the Materials and Methods section to focus on the scientific rationale and key parameters of our analyses, avoiding step-by-step instructions for web interfaces.

Comments 2:

2.         Figures are blurry, with low resolution and unreadable. Need to incorporate good quality images in order to interpret the data and validate the conclusions made.

Response 2: Thank you for your valuable suggestions. In response to your comments, we have comprehensively revised all Figures in the manuscript. And high-resolution versions of these Figures have been prepared and uploaded to the online submission system to ensure they meet the journal's publication standards for clarity and quality.

Comments 3:

3.         It would be nice if the relevant genomic island locations are incorporated in the table S1, so that all relevant data is at one place to access.

Response 3: We appreciate your comment regarding data presentation. The distinct organization of Table 1 (genomic island locations) and Table S1 (resistance gene profiles) was intentionally designed to: (1) maintain clear separation of genomic and functional annotations, (2) enable efficient cross-referencing between island positions and resistance determinants, and (3) facilitate systematic identification of resistance-associated genomic islands through logical data integration. Therefore, we believe that presenting these two sets of results separately has a better logical sequence and can provide an effective basis for the screening of antibiotic resistance-associated genomic islands.

Comments 4:

4.         Discussion part needs to be elaborated to add weightage to the statement 'This finding contributes to our understanding of the mechanisms behind the dissemination of pathogenicity and antibiotic resistance traits in foodborne Proteus vulgaris strains, particularly from the perspective of gene transfer' written in the introduction.

Response 4: We appreciate your valuable suggestions for improving the Discussion section. In response, we have thoroughly revised this section, strengthening the discussion of our findings' implications for food safety and One Health contexts. We believe these modifications have significantly enhanced the section's scientific depth and translational relevance while maintaining clear focus on our core findings.

4. Response to Comments on the Quality of English Language

Point 1: Materials and methods section is written in non-scientific manner and needs improvement.

Response 1: We appreciate your constructive feedback. We have thoroughly implemented comprehensive language refinements throughout the manuscript (including the Materials and Methods section) to enhance scientific rigor, clarity, and readability.

5. Additional clarifications

We sincerely appreciate your thorough review and constructive suggestions, which have greatly enhanced our manuscript. In response to your comments, we have carefully addressed each point, conducted a comprehensive language edit for clarity and precision, and verified all technical content for accuracy. To facilitate your review, we are submitting both a version with tracked changes highlighting all modifications and a clean final version. We are truly grateful for the time and expertise you have dedicated to evaluating our work and would be happy to make any additional revisions if needed.

Round 2

Reviewer 1 Report

Comments and Suggestions for Authors

Abstract

The revised abstract introduces the aim and key findings of the study, which is appreciated. However, the background could be improved to better highlight the novelty and scientific relevance of studying P. vulgaris from a genomic perspective. I recommend briefly clarifying the knowledge gap being addressed and the One Health implications of antimicrobial-resistant P. vulgaris as a foodborne pathogen. This will improve the accessibility and impact of the abstract for readers unfamiliar with the organism.

Introduction

The revised introduction provides a clearer contextual background to the study and outlines the relevance of P. vulgaris as a multidrug-resistant foodborne pathogen more effectively. However, the research objective could be defined more explicitly. I encourage the authors to clarify whether their primary goal is to explore genomic islands in strain P3M or to conduct a more general comparative genomic analysis across different strains. Additionally, the knowledge gap regarding ARG-VFG co-localization within genomic islands should be explicitly stated to emphasize the novelty of the study. Including references that contextualize the role of GIs in the evolution of antimicrobial resistance in other Enterobacterales would further strengthen the introduction

Research Design

The study presents an appropriate and coherent research design for a comparative genomic analysis. The selection of the P. vulgaris P3M strain as a representative, together with the selection of 12 fully sequenced genomes and the application of standard bioinformatics tools, supports the research objective. The revisions have improved clarity and methodological transparency. There are no major concerns regarding the study design.

Materials and Methods

The Methods section has been partially improved, but still lacks key details necessary for reproducibility. Specifically, the manuscript should include: (1) isolation and identification details of the P3M strain (e.g., sample source, culture conditions, confirmatory identification), (2) sequencing and assembly metrics (platform used, coverage, N50, contigs, completeness), and (3) thresholds and parameters used in bioinformatic analyses. Including these aspects would strengthen the scientific rigor of the study.

Results

The Results section has been partially improved in structure and coherence; however, further refinement is needed to enhance clarity and scientific value. Please ensure that all relevant data from the supplemental tables are fully integrated and discussed in the main text. Additionally, avoid redundancy and ensure a more fluid narrative when transitioning between topics (e.g., ARGs, GIs, phylogenetics). Adding quantitative summaries and better contextual interpretation would substantially strengthen the presentation of the results.

Discussion

The discussion has improved in length and breadth, especially in contextualizing the genomic islands and resistance genes. However, it still lacks critical analysis in several areas. I recommend:

  • Expanding comparative insights with similar studies in related Enterobacterales species.
  • Reducing redundancy by avoiding repetition of results already detailed earlier.
  • Strengthening the interpretation of the data, especially regarding the co-occurrence of ARGs and virulence genes, by discussing functional or ecological implications.
  • Reflecting more clearly on the limitations of your study and proposing concrete future directions.

Conclusions

The revised conclusion is clearer and better structured than the original. However, to more effectively reflect the impact of your findings, we suggest:

  • Strengthening the connection between your key genomic results (e.g., types and frequency of resistance/virulence genes in GIs) and their implications.
  • Acknowledging limitations, such as a lack of functional validation or environmental sampling.
  • Including concrete proposals for future work (e.g., transcriptomic analyses or in vivo validation).
    These additions would make the conclusion more persuasive and well-grounded in your data.

Supplementary Tables (1s–3s)

Figures and supplementary tables are informative and generally well structured, but several improvements are needed:

  • Expand figure legends to improve the interpretation and contextualisation of the results.
  • Standardize terminology across all tables and figures (e.g., ARGs vs. resistance genes).
  • Clearly define scoring thresholds or criteria in supplementary tables (e.g., identity %, e-values).
  • Ensure that phylogenetic trees (if present) are properly labeled and include scale bars and confidence values.
  • These refinements will improve the clarity and impact of your visual data presentation.

References

The cited references are largely appropriate, recent, and relevant to the manuscript’s scope. You have included foundational studies on antimicrobial resistance, genomic islands, and comparative genomics. To further strengthen the One Health framing, consider adding a few more references that address Proteus vulgaris as an emerging zoonotic or foodborne pathogen in the context of public health.

Quality of English Language

While the English language has been improved compared to the initial version, some sections still contain awkward or unclear phrasing that may obscure the scientific message. I recommend a thorough language revision by a native English speaker or professional editing service to enhance clarity, improve sentence flow, and ensure scientific precision throughout the manuscript.

Comments on the Quality of English Language

Quality of English Language

Although the manuscript has been revised and improved with respect to the previous version, there are still confusing grammatical constructions, imprecise use of verb tenses, and long sentences that affect the clarity of the message. This does not impede the understanding of the content, but it does make fluent reading difficult and reduces the scientific impact of the text. In addition, there are sentences translated literally from Chinese or non-native structures that require correction. I recommend a thorough language revision by a native English speaker or professional editing service to enhance clarity, improve sentence flow, and ensure scientific precision throughout the manuscript.

🔹 Specific areas where English can be improved:

  • Abstract and Introduction: some sentences are unnecessarily long or present a syntactic disorder.
  • Materials and Methods: the passive voice predominates, and clarity is lost in the methodological sequence.
  • Discussion and Conclusion: there are sentences with semantic ambiguity or constructions with poor connectors ("this is" and "that means").

Reviewer 2 Report

Comments and Suggestions for Authors

I would like to commend the authors for the substantial improvements made to this manuscript. The quality and scientific rigor have been considerably enhanced, addressing most of the previously identified concerns.

The methodological descriptions are now appropriate, the terminology has been corrected, and the overall presentation has improved significantly. The following suggestions are intended to further refine the manuscript: 

1. The manuscript lacks  information regarding the 13 Proteus vulgaris strains analyzed. The inclusion of a comprehensive supplementary table that includes: strain name, accession code, isolation source, geographic location, collection date, sequencing technology, genome coverage, assembly statistics, and quality metrics could improve the manuscript. Additionally, please clarify whether these represent all complete P. vulgaris genomes available in public databases or specify the selection criteria used.

2. P. vulgaris  appear without proper italicization page 2. Please ensure consistent formatting of all bacterial species names according to standard taxonomical conventions.

3. The CARD database URL appears duplicated on 2.3 section

4. The phylogenetic analysis description requires improved  flow for better readability.

5. The threshold parameters (≥80% identity/≥80% coverage for ARGs; ≥70% identity/≥50% coverage for virulence genes) lack supporting references. Please cite established studies that validate these cutoffs or provide methodological justification for their selection.

6. Sections 3.1 and 3.2 both discuss the 218 identified ARGs, creating unnecessary redundancy. Consider restructuring to eliminate this duplication

7. The figures in the submitted manuscript appear with insufficient resolution for proper evaluation. The images are of low quality and were not provided as separate high-resolution files, making it difficult to assess their clarity, readability, and scientific accuracy.

Reviewer 3 Report

Comments and Suggestions for Authors

The revised manuscript demonstrates substantial improvement. All the concerns mentioned earlier have been effectively addressed and overall quality of the work has been improved greatly. The structure, clarity, and presentation of the content are now much stronger, resulting in a more coherent and compelling manuscript. 
